# REMOTE REINFORCEMENT LEARNING WITH COMMUNICATION CONSTRAINTS

## ABSTRACT

We introduce the novel problem of *remote reinforcement learning (RRL) with a communication constraint*, in which the actor that takes the actions in the environment lacks direct access to the reward signal. Instead, the rewards are observed by a controller, which communicates with the agent through a communication-constrained channel. This can model a remote control scenario over a wireless channel, where the communication link from the controller to the agent has limited capacity due to power, bandwidth, or delay constraints. In the proposed solution, rather than transmitting the reward values to the agent over the rate-limited channel, the controller learns the optimal policy, and at each round, signals the action that the agent should take over the channel. However, instead of sending the precise action–which can be prohibitive when the action set is large–we use an importance sampling approach to reduce the communication load, which allows the agent to sample an action from the current policy. The actor, sampling from the desired policy at each turn, can also learn the optimal policy, albeit at a slower pace, using supervised learning. We exploit the learned policy at the actor to further reduce the communication load. Our solution, called Guided Remote Action Sampling Policy (GRASP), exhibits a significant reduction in communication requirements, achieving an average of 12-fold decrease in data transmission across all experiments, and 50-fold reduction for environments with continuous action spaces. We also show the applicability of GRASP beyond single-agent scenarios, including parallel and multi-agent environments.

## 1 INTRODUCTION

Reinforcement learning (RL) enables the solution of complex, sequential tasks through interaction with the environment alone. This is accomplished by identifying a sequence of actions that maximize the cumulative expected rewards. However, the reward signal is not always readily available to the agent, as it can be difficult to evaluate or costly to acquire. For instance, in human-in-the-loop systems, the reward may need to be evaluated and provided by a human, which can cause delays (Knox & Stone, 2009; Daniel et al., 2014), or be learned from demonstrations (Abbeel & Ng, 2004; Schaal, 1996; Arora & Doshi, 2021). In other complex engineering systems, such as communication networks or multi-processor systems, evaluating the reward may require solving complex optimization problems or accumulating information distributed across a large network. The challenges of lacking or costly reward acquisition in RL have been studied in the context of active learning (Krueger et al., 2020; Eberhard et al., 2024).

In this work, we consider a distributed learning scenario with two agents: a controller and an actor. Only the controller has access to the reward signal, while the actor takes the actions. This setting is depicted in Figure 1. The actor observes the state of the environment, either fully or partially, and decides on an action; however, it does not have access to the reward. The controller observes both the state of the environment and the reward signal, but relies on the actor to take actions. The controller communicates with the actor over a rate-limited channel to help guide it toward the correct action. We dub this problem *remote reinforcement learning (RRL) with a communication constraint*.

If the controller is able to convey the reward signal to the actor through the communication channel, the actor would have all the necessary information to perform RL; that is, it could learn a policy that probabilistically maps states to actions to maximize the sum of future rewards. However, this

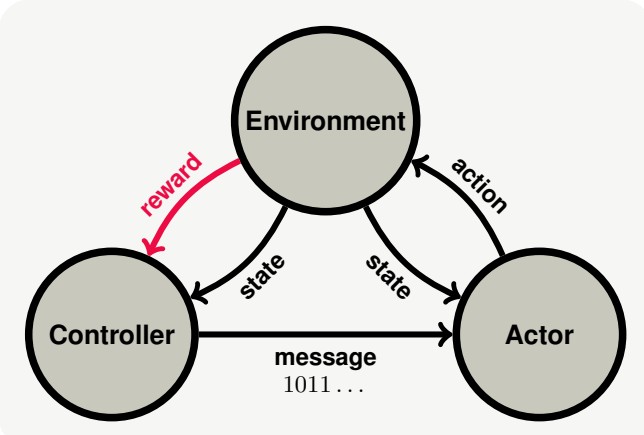

Figure 1: Illustration of the RRL problem. Both the actor and the controller observe the state of the environment. The controller sends a message to the actor over a constrained communication channel, and the actor selects the appropriate action. Only the controller receives the reward signal.

approach encounters four primary limitations in our scenario: limited communication, feasibility, parallelism, and coordination. Firstly, the reward is usually a real number, and it may not be possible to represent it exactly with the finite number of bits dictated by the capacity of the communication channel between the controller and the actor. Secondly, the actor may be deployed on a resource-constrained environment, such as an edge devices or a sensor, and may not be capable of running a complex RL algorithm locally, even if it has full or partial access to the reward signal. Thirdly, to accelerate learning, multiple concurrent agents are often used to collect experiences independently (Mnih et al., 2016; Heess et al., 2017). In our framework, this corresponds to communicating with multiple actors, each interacting with the same but parallel environments. However, if the actors received individual reward signals, they would develop distinct policies, failing to benefit from shared experiences. Lastly, in multi-agent reinforcement learning (MARL) scenarios, where multiple actors jointly influence the same environment, simply conveying individual reward signals would result in a distributed training algorithm that struggles with action coordination. These limitations suggest that direct communication of the reward signal is an inefficient solution for RRL.

Shifting the focus to the controller, it has full knowledge of the state and rewards. If it also had access to the actions, it could effectively run a RL algorithm locally to obtain the optimal policy. This would emulate the best possible performance of a centralized learning scenario, provided the controller can select and communicate the subsequent actions to the actor at each decision step. In scenarios involving small discrete action spaces, this method can result in smaller message sizes compared to conveying the reward signal (or a quantized version of it) to the actor. On the other hand, for continuous action spaces, one might initially think that communicating actions would face similar bandwidth limitations as with reward transmissions, given that actions in such spaces can assume an uncountably infinite number of values, necessitating some form of quantization and compression. However, crucially, the actor does not need to take a specific action from the controller's policy, but any sample from it would suffice. Let $P$ be the distribution of actions dictated by the controller's policy in a given state, while $Q$ represents the actor's belief about the policy in this state. From an information-theoretic perspective, the number of bits required to communicate a particular sample from $P$ (i.e., a specific action) is approximately $\mathbf{H}(P) + \mathbf{D}_{KL}[P||Q]$—the entropy of the action plus the cost of using the 'wrong' distribution $Q$ to compress it. Instead, by generating candidate samples from $Q$ and using $P$ only to select a single candidate via an importance-sampling-like criterion, the cost of communicating the index of the accepted sample can be reduced to approximately $\mathbf{D}_{KL}[P||Q]$ (Cuff, 2008; Li & El Gamal, 2018). This method of conveying random actions is particularly effective in systems with multiple parallel agents. By centrally processing all collected experiences, the controller can learn the most informed policy, benefiting from the experiences of all the actors in parallel. The controller can then enable each actor to take an action based on the most up-to-date policy in the next round. We call this approach the Guided Remote Action Sampling Policy (GRASP) method.

MARL extends the traditional RL framework to multiple agents, where the agents collectively influence the environment's state. This scenario is particularly relevant to RRL because the reward is often tied to the overall system's performance; and thus, may not be directly accessible to each actor. Moreover, decentralized MARL suffers from a high degree of non-stationarity (Du & Ding, 2021; Wong et al., 2023). If each agent views others as part of the environment, the learning and policy updates by other agents alter the environment, rendering it highly non-stationary and challenging to learn from. To address this issue, a centralized-learning decentralized-execution approach is typically employed (Lowe et al., 2017). During training, this method involves centrally learning the policies of all agents using global information, thereby avoiding the non-stationarity problem. After training, these policies are fixed, ensuring that even though the agents execute them independently, the environment remains consistent for each agent. Multiple agents in MARL translate into multiple actors in RRL, while a single centralized controller is ideally suited to oversee the centralized training stage, enabling the actors to take correlated actions at each step.

The remainder of the paper is organized as follows: Section 2 provides the background and reviews related works. Section 3 mathematically defines the framework for RRL. Section 4 empirically evaluates the proposed approach, comparing it against other solutions. Finally, the paper concludes with a summary of findings and proposes potential future research directions.

The logarithms are base 2, $\mathbb{E}\left[\cdot\right]$ denotes expectation, $\mathbf{H}\left(P\right) \triangleq \mathbb{E}_{X \sim P}\left[\log p(x)\right]$ represents the entropy of a random variable distributed according to $P$, or differential entropy in the case of continuous random variables, and $\mathbf{D}_{KL}\left[P||Q\right] \triangleq \mathbb{E}_{x \sim P}\left[\log \frac{p(x)}{q(x)}\right]$ denotes the Kullback-Leibler divergence between distributions $P$ and $Q$.

## 2 BACKGROUND AND RELATED WORKS

### 2.1 RL WITH COMMUNICATION CONSTRAINTS

RL literature includes many connections to communications. Relevant works include federated RL (Nadiger et al., 2019; Jin et al., 2022), where multiple agents collaborate to learn a common policy while keeping data localized to each agent. This contrasts with RRL, where both the controller and the actors have access to the state and the actions. MARL with communication among agents is an extensively studied topic, where the agents exchange messages over a dedicated link (Foerster et al., 2016; Wang et al., 2020), including over noisy channels (Tung et al., 2021; Roig & Gündüz, 2020), to achieve a common goal. In these works, reward is known to all the actor(s), unlike in our setting, where it is only accessible to a remote controller. Our work is orthogonal to these approaches; GRASP can be applied to solve MARL problems through centralized training, with or without communication between actors. In the presence of communication, an agent's messages can be considered as part of its action space; thus, during training, they would be chosen by and known to the controller. Furthermore, in this scenario, the centralized training with decentralized execution paradigm is often employed, to which GRASP is particularly well-suited.

Communication constraints have also been recently considered for distributed multi-armed bandit problems in Hanna et al. (2022); Mitra et al. (2023); Salgia & Zhao (2023). These works focus on the compression of the reward signal, or the model, to minimize regret. Differently from our setting, in these papers, the agents taking the actions observe the corresponding reward, which they then report to the learning agent over a limited channel. The work closest to ours is Pase et al. (2022), which studies sending actions over a communication-limited channel in a contextual multi-armed bandit problem. In contrast to our work, the states are independent across time, and the agents cannot learn the policy. The authors study the regret behavior for a certain class of policies, focusing on the asymptotic regime of infinitely many agents.

### 2.2 REMOTE SAMPLING

Let $P$ be the distribution we wish to sample from in RRL, representing the controller's policy, and $Q$ be a reference distribution, representing the actor's policy, which is also known to the controller. Our goal is to enable the agent to sample from $P$ by communicating as few bits as possible. This remote sampling problem is also known as 'reverse channel coding' (Bennett et al., 2002) or 'channel simulation' (Cuff, 2008) in the literature. In their quest to obtain the entanglement-assisted capacity

of a quantum channel, Bennett *et al.* proved the asymptotic equivalence of all discrete communication channels of equal capacity, that is, they can simulate each other in the presence of sufficient common randomness. Cuff (2008) studied the asymptotic per-symbol rate, focusing on the impact of limited common randomness between the encoder and the decoder. Traditionally, channel simulation has been studied in a slightly different setting, in which for a given joint distribution $P_{XZ}$, the encoder first samples $z \sim P_Z$, and the decoder aims to sample from $P_{X|Z=z}$, which represents the target distribution $P$. The reference distribution $Q = P_X$ is the marginal distribution over all values of $Z$. The results are then given in terms of the mutual information between $X$ and $Z$, i.e., $I(X, Z) = \mathbb{E}_{z \sim P_Z} \left[ \mathbf{D}_{KL} \left[ P_{X|Z=z} || P_X \right] \right]$, the expected KL-divergence. However, as noted in Theis & Yosri (2022), we can translate between these two viewpoints, and the relevant results apply to the version used for RRL in this paper.

A naive approach to this problem would be to sample an action from $P$ at the controller and send it using universal lossless data compression. The advantage of the channel simulation approach over directly sampling from $P$ has been shown in Li & El Gamal (2018); Theis & Yosri (2022): the number of bits required to communicate the index of the selected candidate sample is approximately $\mathbf{D}_{KL} [P||Q]$, whereas directly communicating the action requires at least $\mathbf{H} [P] + \mathbf{D}_{KL} [P||Q]$ bits. Importantly, this approach allows for communicating samples from a continuous distribution $P$ by transmitting a finite number of bits, provided that $\mathbf{D}_{KL} [P||Q] < \infty$.

One-shot results, focusing on sending a single sample, were obtained for discrete distributions in Harsha et al. (2010), and later improved and generalized to continuous random variables using functional representation in Li & El Gamal (2018). The current best-known upper bound was derived in Li & Anantharam (2021), demonstrating that the expected message size need not exceed

$$\mathbf{D}_{KL} [P||Q] + \log \left( \mathbf{D}_{KL} [P||Q] + 1 \right) + 4.732 \text{ bits}, \tag{1}$$

which is close to optimal, and follows the following lower bound derived in Li & El Gamal (2018):

$$\mathbf{D}_{KL} [P||Q] + \log \left( \mathbf{D}_{KL} [P||Q] + 1 \right) - 1 \text{ bits}. \tag{2}$$

The importance sampling approach in Harsha et al. (2010) results in a suboptimal rate but provides approximation guarantees when we impose constraints on the computation complexity. These guarantees are achieved by ordered random coding (Theis & Yosri, 2022), which maintains the communication rate of Poisson functional representation.

### 2.3 IMITATION LEARNING

In the proposed solution to the RRL problem, actions need to be effectively communicated from the controller to the actor. To facilitate this, we use channel simulation, which enables the transmission of actions using approximately $\mathbf{D}_{KL} [P||Q]$ bits, where $P$ represents the action probability distribution under the controller's policy in a given state, and $Q$ is a probability distribution known to both the controller and the actor. What should $Q$ be? One solution is to periodically transmit the controller's current policy to the actor and use it as the reference distribution $Q$. This method involves resending updates to account for the evolving policy as the controller learns. Since the policies are represented as neural networks, this approach requires periodically transmitting all the parameters, which is very costly from a communication perspective.

Alternatively, since the actor can observe the current state and receives a sample from the desired policy, it can learn the controller's policy—-a probability distribution over actions conditioned on the state—-in a supervised manner. This concept is known as *behavioral cloning* and is an application within imitation learning, a field focused on learning policies from demonstrations (Pomerleau, 1988; Torabi et al., 2018; Abbeel & Ng, 2004; Schaal, 1996; Arora & Doshi, 2021). Inverse RL (Arora & Doshi, 2021) offers another approach, where the objective is to recover the reward function from a set of state-action trajectories. While this approach can succeed in scenarios where behavioral cloning fails, it is also more complex, often requiring the solution of RL problems as a subroutine. A combination of these two approaches was proposed by Ho & Ermon (2016), where a policy is learned directly as if learning from rewards recovered through inverse RL, without explicitly solving the inverse problem. In our experiments, we found that behavioral cloning alone was sufficient for our purposes, and we provide a more thorough examination of this in Section 4.

---

**Algorithm 1** GRASP Controller

---

**Require:** Initial controller policy parameters $\theta$, initial actor policy parameters $\phi$
1: **for** $epoch = 0$ **to** $T$/batch_size **do**
2:    **for** $step = 0$ **to** batch_size **do**
3:       $t \leftarrow epoch \times$ batch_size $+ step$
4:       $s_t \leftarrow$ observe state from environment
5:       $P \leftarrow$ action distribution under controllers policy$(s_t, \theta)$
6:       $Q \leftarrow$ action distribution under actors policy$(s_t, \phi)$
7:       $a_t, m_t \leftarrow$ channel simulation encoding$(P, Q)$
8:       Send $m_t$ to actor
9:       $r_t \leftarrow$ reward from environment
10:    **end for**
11:    $b \leftarrow epoch \times$ batch_size
12:    $e \leftarrow b +$ batch_size
13:    Update $\theta$ based on $s_{[b:e]}, a_{[b:e]}, r_{[b:e]}$ using online RL
14:    Update $\phi$ based on $s_{[b:e]}, a_{[b:e]}$ using supervised learning
15: **end for**

---

**Algorithm 2** GRASP Actor

---

**Require:** Initial actor policy parameters $\phi$
1: **for** $epoch = 0$ **to** $T$/batch_size **do**
2:    **for** $step = 0$ **to** batch_size **do**
3:       $t \leftarrow epoch \times$ batch_size $+ step$
4:       $s_t \leftarrow$ observe state of the environment
5:       $Q \leftarrow$ action distribution under actors policy$(s_t, \phi)$
6:       $m_t \leftarrow$ receive message from the controller
7:       $a_t \leftarrow$ channel simulation decoding$(mes_t, Q)$
8:       act in environment$(a_t)$
9:    **end for**
10:    $b \leftarrow epoch \times$ batch_size
11:    $e \leftarrow b +$ batch_size
12:    Update $\phi$ based on $s_{[b:e]}, a_{[b:e]}$ using supervised learning
13: **end for**

---

## 3 REMOTE REINFORCEMENT LEARNING (RRL)

In this section, we formally define the RRL problem. For simplicity of notation, we focus on the single-actor case in this work, but the extension to multiple actors follows similar mathematical arguments. Any Markov decision process can be converted into an RRL problem; it is described by a tuple $M = (S, s_0, A, p_T, R, \gamma)$, where $S$ is the set of states, $s_0$ is the initial state, $A$ is the set of actions, $p_T(s'|s, a) : S \times A \rightarrow \mathcal{P}(S)$ represents the transition probability of moving to the subsequent state $s'$ given the current state $s$ and action $a$, $R(s_t, s_{t+1}, a) : S^2 \times A \rightarrow \mathcal{P}(\mathbb{R})$ is the reward function, and $\gamma \in [0, 1)$ is the discount factor (Sutton & Barto, 1998). The objective is to find a policy $\pi : S \rightarrow \mathcal{P}(A)$ that maximizes the sum of discounted rewards:

$$\pi^* = \arg\max_{\pi} \sum_{t=0}^{\infty} \gamma^t \mathbb{E}_{\substack{a_t \sim \pi(s_t) \\ s_{t+1} \sim p_T(s_{t+1}|s_t, a_t) \\ r_i \sim R(s_t, s_{t+1}, a_t)}} [r_i] . \tag{3}$$

At each time step, the current state $s_t$ is observed by both the controller and the actor. The controller transmits a variable-length message $m_i = f(s_{[:t]}, r_{[:t-1]})$, for some encoding function $f : S^t \times \mathbb{R}^{t-1} \rightarrow \{0, 1\}^*$, based on all the states and rewards observed so far. The actor then chooses an action $a_t = g(s_{[:t]}, a_{[:t-1]}, m_{[:t]})$ using a function $g : S^t \times A^{t-1} \times (\{0, 1\}^*)^t \rightarrow A$.

The pseudocode for the proposed GRASP method, as outlined in Sections 1 and 2, is provided in Algorithm 1 for the controller and in Algorithm 2 for the actor. The controller maintains a copy of the actor's parameters because, to use channel simulation, both parties (the encoder and the decoder) need access to a common distribution $Q$. In GRASP, we employ the actor's current policy

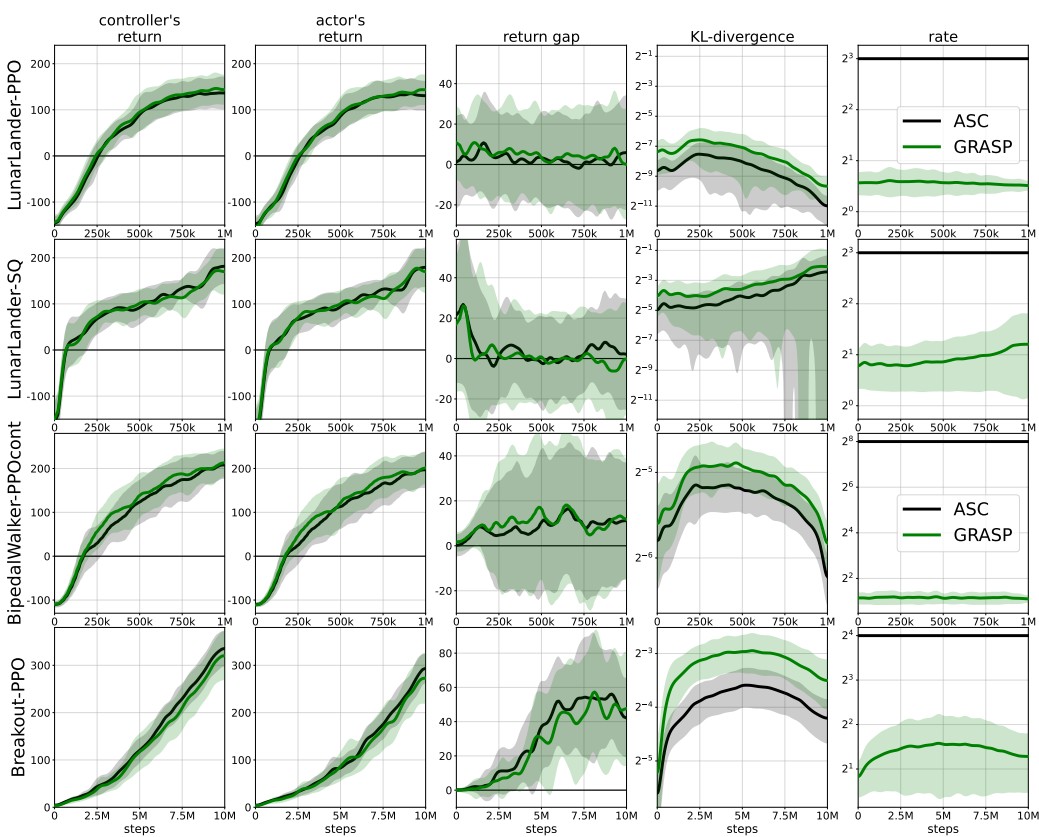

Figure 2: Training plots for different single-agent RL environments in the RRL setting, comparing sending actions with source coding (labeled ASC) and GRASP (using channel simulation to communicate actions combined with behavioral cloning). The algorithms used include PPO, continuous PPO, and soft Q-learning (SQ). The thick lines indicate the mean, while the shaded regions represent the standard deviation. For readability, the values are smoothed with a Gaussian kernel with a standard deviation equal to 2% of the number of training steps for each environment.

conditioned on the current state as the common distribution. This policy is never enacted; that is, the actor's actions do not follow it directly but are instead used solely to facilitate efficient communication of actions derived from the controller's policy. Additionally, the parameters of the actor's network are never explicitly communicated; they are updated based on the observed actions and states, allowing them to evolve in lockstep between the actor and the controller. In particular, to minimize the communication cost, we need to minimize the KL-divergence between the controller's policy $\pi_C$ and the actor's policy $\pi_A$, which corresponds to minimizing the empirical cross-entropy:

$$\arg\min \mathbb{E}_s \left[ \mathbf{D}_{KL} \left[ \pi_C(\cdot|s) || \pi_A(\cdot|s) \right] \right] \simeq \arg\min \frac{1}{N} \sum_{i=1}^{N} -\log \pi_A(a_i|s_i)$$

where the expectation over states is based on the policy $\pi_C$, and $a_i, s_i, i \in \{1, 2, \ldots N\}$ are the observed actions and states.

## 4 EXPERIMENTS

The two main claims of our work are that GRASP does not negatively impact training and that it leads to significant communication savings. To evaluate its effectiveness, we assess it across a range of RL environments. We compare GRASP against two benchmarks. In the first benchmark, the controller decides which action to take at each time step and communicates this action to the actor using

Table 1: Performance of GRASP and ASC in various RRL environments

| environment | algorithm | training method | controller final return | | actor final return | | return gap | | norm. return gap (%) | |
|---|---|---|---|---|---|---|---|---|---|---|
| LunarLander | PPO | ASC | 135 | (31) | 130 | (27) | 5.1 | (14.2) | 1.7 | (4.7) |
| | | GRASP | 141 | (29) | 142 | (28) | -0.9 | (16.9) | -0.3 | (5.5) |
| LunarLander | SQ | ASC | 180 | (35) | 178 | (44) | 1.3 | (23.8) | 0.3 | (6.2) |
| | | GRASP | 169 | (44) | 169 | (39) | -0.5 | (21.4) | -0.1 | (5.7) |
| BipedalWalker | PPOcont | ASC | 209 | (28) | 196 | (38) | 12.6 | (17.8) | 3.9 | (5.6) |
| | | GRASP | 214 | (30) | 205 | (33) | 9.6 | (15.2) | 2.9 | (4.7) |
| Breakout | PPO | ASC | 340 | (38) | 299 | (29) | 41.4 | (24.0) | 12.3 | (7.1) |
| | | GRASP | 323 | (49) | 274 | (57) | 48.8 | (29.4) | 15.2 | (9.2) |
| CooperativePong | PPO | ASC | 87 | (6) | 85 | (5) | 2.1 | (6.2) | 2.6 | (7.7) |
| | | GRASP | 84 | (5) | 80 | (4) | 3.9 | (3.9) | 5.0 | (5.1) |
| PistonBall | PPOcont | ASC | 92 | (3) | 85 | (10) | 6.8 | (9.5) | 7.3 | (10.1) |
| | | GRASP | 91 | (3) | 85 | (11) | 5.8 | (11.0) | 6.0 | (11.4) |
| Spread | PPOcont | ASC | -30 | (1) | -30 | (1) | -0.1 | (0.8) | -1.9 | (12.2) |
| | | GRASP | -30 | (1) | -30 | (1) | -0.3 | (0.8) | -4.4 | (12.6) |

source coding, referred to as ASC. The second benchmark involves transmitting the reward directly to the actor. In our implementation, we assume that the reward at each time step is sent using 32 bits. It is also possible to consider further quantization of the reward signal, though this may come at the cost of reduced performance. GRASP is compatible with any RL algorithm. For our experiments, we focused on proximal policy optimization (PPO) (Schulman et al., 2017), a de-facto standard in RL. Additionally, we applied it to other algorithms such as deep Q-learning (DQN) (Mnih et al., 2013), soft Q-learning (SQ) (Haarnoja et al., 2017), and deep deterministic policy gradients (DDPG) (Lillicrap et al., 2016). We employ the CleanRL open-source library implementation (Huang et al., 2022), using the default hyperparameters, if present, for each environment. These include neural network architecture, learning rate, and other algorithm-specific settings, with the full list provided in Appendix C. GRASP also entails learning the actor's policy in a behavioral cloning manner. For the actor, we utilize the same hyperparameters and architecture as the controller, training the policy using cross-entropy loss. For the channel simulation method, we opted for ordered random coding (Theis & Yosri, 2022). To ensure a comprehensive evaluation, we selected a diverse set of environments that vary in difficulty, type of action spaces (discrete and continuous), type of observations (fully and partially observable, proprioceptive, and image-based), as well as with single and multiple agents. These environments include CartPole and Pendulum from Classic Control, LunarLander and BipedalWalker from Box2D, HalfCheetah from MuJoCo, the Atari game Breakout, which were simulated using the Gymnasium library (Towers et al., 2023), and CooperativePong and PistonBall from the PettingZoo library (Terry et al., 2021). The experiments were repeated across 20 independent and seeded runs, except for Breakout and CooperativePong, which were performed 8 times; all reported values are averaged and include the standard deviation.

The single-agent training progress plots are presented in Figure 2, comparing GRASP with directly sending the controller's actions without channel simulation. The first column describes the controller's return throughout training; every 10 000 steps, the controller's policy was evaluated across 30 episodes, recording the mean sum of rewards. The training performance is consistent between the two approaches in all environments. The final returns of the controller are reported in Table 4 with standard deviations, showing that the two approaches learn equally effective policies. The second column in Figure 2 depicts the return of the actor's policy; that is, a policy learned through supervised learning (behavioral cloning) by the actor based on the actions communicated by the controller. It is evaluated in the same manner as the controller's policy. As previously mentioned, this policy is not followed during training, but is used in channel simulation to reduce the communication cost. It is not used for the ASC variant during training. In both cases, we observe that the training trajectories resemble that of the controller's policy-—the actor learns a useful policy

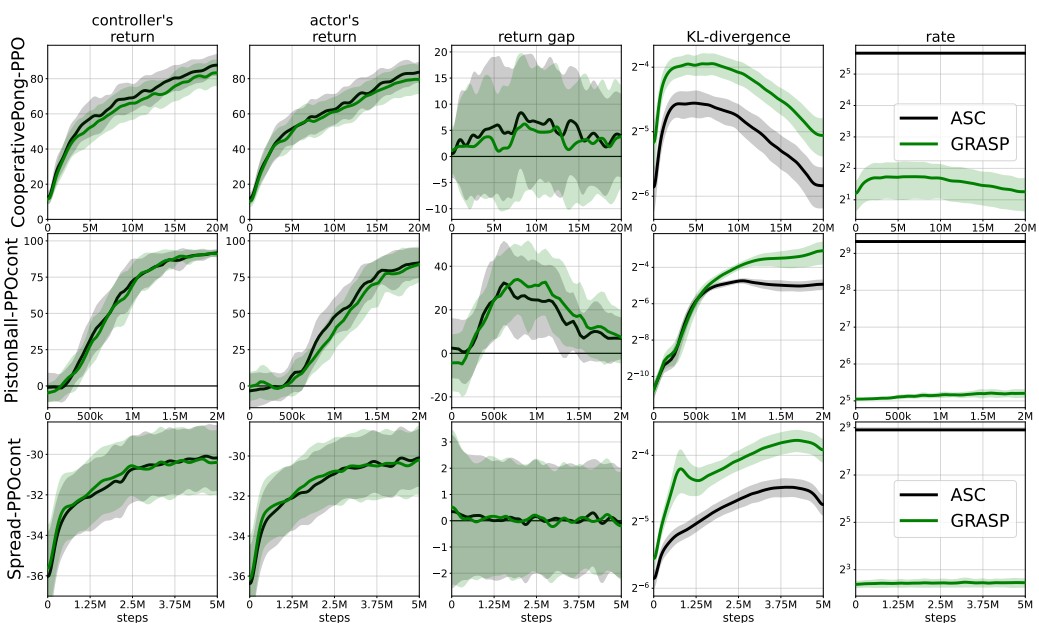

Figure 3: Training plots for different multi-agent RL environments in the RRL setting.

through behavioral cloning. Depending on the use case, after training, the controller might transmit its learned policy to the actor, or if the actor's policy is adequate, no further communication is necessary. The difference between the final performance of the controller's and actor's policies is shown in the third column of Table 4. The next column describes the gap in normalized terms according to the formula $\frac{\text{return gap}}{\text{average final return}-\text{random policy return}}\%$. Except for the Breakout environment, the final performance of the actor's policy is within a few percentage points of the controller's, demonstrating that behavioral cloning can serve as an effective alternative to policy transmission. In the LunarLander environment, we observe that GRASP achieves the same results as ASC when using PPO and soft Q-learning. This trend holds across other experiments, detailed in Appendix A, where we also evaluate the performance of the DQN algorithm. Further experiments in the appendix include Cart-Pole with PPO, DQN, and SoftDQN, and Pendulum and HalfCheetah with PPOCont and DDPG, confirming that GRASP remains robust across a variety of environments and RL algorithms.

The training plots for multi-agent environments are shown in Figure 3, following the same methodology. To further compare different scenarios, we allow both agents in CooperativePong to share the same policy. While in PistonBall and Spread, only the controller is centralized, and each of the actors—20 in PistonBall and 3 in Spread—learns its own policy. As in previous experiments, we observe that GRASP and ASC achieve similar performance.

The communication cost of the considered alternatives are plotted in the last column of Figures 2 and 3. For ASC, the cost of sending discrete actions is calculated as the logarithm of the cardinality of the action set, while for continuous spaces, we followed the environments' specifications, which require 32-bit floats per action dimension. For GRASP, we used ordered random coding to communicate samples from the controller's policy, and calculated the log probability of the selected index as the communication cost. We observe that GRASP consistently outperforms ASC, often by many orders of magnitude. The total communication costs are outlined in Table 4, where GRASP offers between 4.2- and 115-fold communication savings compared to ASC, with a geometric average of 13 times reduction. The most significant savings are observed in environments with continuous actions. Sending the reward is functionally equivalent to action source coding, with the key difference being that only the actor's model is trained. Therefore, the difference between the two is where the intelligence—-and thus, the computational complexity-—will be placed. Assuming a communication rate of 32 bits per time step, GRASP achieves communication savings ranging from 6.3- to 343-fold, with a geometric average of 41 times less communication than sending the reward.

Table 2: Communication rate of GRASP and ASC across RRL environments

| environment | algorithm | training method | mean KL-div | total # of communicated bits | rate reduction |
|---|---|---|---|---|---|
| LunarLander | PPO | ASC | 0.003 (0.000) | 1.91Mb (0b) | |
| | | GRASP | 0.006 (0.000) | 361.10Kb (1.31Kb) | ×5.41 |
| LunarLander | SQ | ASC | 0.074 (0.005) | 1.91Mb (0b) | |
| | | GRASP | 0.109 (0.012) | 463.37Kb (10.17Kb) | ×4.22 |
| BipedalWalker | PPOcont | ASC | 0.024 (0.001) | 122.07Mb (0b) | |
| | | GRASP | 0.029 (0.001) | 1.06Mb (10.28Kb) | ×114.89 |
| Breakout | PPO | ASC | 0.067 (0.010) | 19.07Mb (0b) | |
| | | GRASP | 0.109 (0.019) | 3.21Mb (1.02Mb) | ×5.95 |
| CooperativePong | PPO | ASC | 0.032 (0.003) | 30.23Mb (0b) | |
| | | GRASP | 0.052 (0.001) | 1.78Mb (24.42Kb) | ×17.01 |
| PistonBall | PPOcont | ASC | 0.025 (0.002) | 61.04Mb (0b) | |
| | | GRASP | 0.057 (0.012) | 3.38Mb (41.76Kb) | ×18.08 |
| Spread | PPOcont | ASC | 0.037 (0.002) | 762.94Mb (0b) | |
| | | GRASP | 0.058 (0.006) | 8.60Mb (52.44Kb) | ×88.72 |

## 5 LIMITATIONS

To perform channel simulation, both parties require access to a common reference distribution $Q$. In GRASP, this is achieved by training an additional policy at the actor, which aims to follow the controller's policy as closely as possible. The closer the two policies are, the smaller the communication cost. This requirement introduces increased computational cost at the actor in order to reduce the communication rate. As previously mentioned, the need for a common distribution $Q$ can be circumvented by periodically transmitting the controller's current policy to the actor. This approach can reduce the need for training a separate policy at the actor, but it may lead to periodic spikes in communication load, depending on the frequency and size of the transmitted policy updates.

RRL assumes that both the agent and the controller have access to the same state/observation. In situations where this is not the case, a common policy cannot be trained, and thus GRASP cannot be implemented. However, there exists a potential avenue due to recent advances in the information theory literature regarding the error rates of performing channel simulation when the encoder and decoder do not share the same policies (Li & Anantharam, 2021). It remains to be determined how best to exploit the different information available to the controller and the actor in such situations to find a good policy in a computation- and communication-efficient manner.

## 6 CONCLUSION

In this work, we have introduced the novel problem of RRL, in which the reward signal is only available to a *controller*, removed from the action-choosing agent, called the *actor*. The actor relies on messages transmitted by the controller to decide on its actions. There are two obvious benchmarks: In the first, the controller conveys the reward signal to the actor, so that the actor can learn the optimal policy by applying its favourite RL algorithm. In the second, the controller learns the optimal policy and transmits the optimal action to the actor at each step. Both of these options may become infeasible when the reward function takes real values or when the action set is prohibitively large (even continuous). We have proposed a novel alternative method, called GRASP, based on importance sampling and behavioral cloning. The controller sends a sample from the desired policy to the actor, and to further reduce the communication cost, the actor attempts to estimate the controller's policy through supervised learning. Our experiments have shown that the proposed method vastly outperforms the baselines, achieving a 12-fold reduction in communication rate while maintaining the same reward.

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

# A  ADDITIONAL RESULTS

In this appendix we include other experiments mentioned in the main text. The training plots are depicted in Figure 4, the end performance is shown in Table 3, and the rate is presented in Table 4.

Table 3: Performance of GRASP and ASC in various RRL environments

| environment | algorithm | training method | controller final return | actor final return | return gap | norm. return gap (%) |
|---|---|---|---|---|---|---|
| CartPole | PPO | ASC | 500 (0) | 500 (0) | 0.0 (0.0) | 0.0 (0.0) |
| | | GRASP | 500 (0) | 500 (0) | 0.0 (0.0) | 0.0 (0.0) |
| CartPole | DQN | ASC | 415 (95) | 432 (81) | -16.7 (57.8) | -4.3 (14.7) |
| | | GRASP | 475 (40) | 458 (53) | 16.4 (43.1) | 3.6 (9.5) |
| CartPole | SQ | ASC | 481 (48) | 463 (68) | 17.4 (50.8) | 3.8 (11.1) |
| | | GRASP | 468 (77) | 463 (79) | 4.7 (14.6) | 1.1 (3.3) |
| Pendulum | PPOcont | ASC | -153 (20) | -154 (21) | 1.9 (4.8) | 0.2 (0.5) |
| | | GRASP | -153 (20) | -155 (22) | 1.9 (7.8) | 0.2 (0.7) |
| Pendulum | DDPG | ASC | -157 (28) | -246 (136) | 89.7 (126.1) | 7.4 (10.4) |
| | | GRASP | -156 (23) | -191 (81) | 35.4 (68.2) | 2.9 (5.6) |
| LunarLander | DQN | ASC | 234 (22) | 207 (37) | 27.4 (29.2) | 6.6 (7.1) |
| | | GRASP | 215 (33) | 190 (30) | 25.2 (30.2) | 6.4 (7.7) |
| HalfCheetah | PPOcont | ASC | 1084 (251) | 1020 (233) | 63.4 (54.9) | 4.4 (3.8) |
| | | GRASP | 1058 (277) | 977 (253) | 81.4 (52.2) | 5.8 (3.7) |
| HalfCheetah | DDPG | ASC | 4662 (1429) | 3716 (1776) | 945.9 (1132.1) | 20.2 (24.2) |
| | | GRASP | 4113 (1449) | 3765 (1642) | 348.7 (766.1) | 8.4 (18.6) |

Table 4: Communication rate of GRASP and ASC across RRL environments

| environment | algorithm | training method | mean KL-div | total # of communicated bits | rate reduction |
|---|---|---|---|---|---|
| CartPole | PPO | ASC | 0.005 (0.000) | 488.28Kb (0b) | |
| | | GRASP | 0.020 (0.002) | 188.81Kb (3.84Kb) | ×2.59 |
| CartPole | DQN | ASC | 0.257 (0.018) | 488.28Kb (0b) | |
| | | GRASP | 0.269 (0.025) | 305.06Kb (13.29Kb) | ×1.60 |
| CartPole | SQ | ASC | 0.059 (0.013) | 488.28Kb (0b) | |
| | | GRASP | 0.094 (0.023) | 219.47Kb (10.15Kb) | ×2.22 |
| Pendulum | PPOcont | ASC | 0.009 (0.001) | 15.26Mb (0b) | |
| | | GRASP | 0.011 (0.001) | 524.25Kb (6.86Kb) | ×29.80 |
| Pendulum | DDPG | ASC | 4.952 (0.932) | 15.26Mb (0b) | |
| | | GRASP | 5.193 (0.993) | 1.47Mb (108.43Kb) | ×10.37 |
| LunarLander | DQN | ASC | 0.536 (0.020) | 1.91Mb (0b) | |
| | | GRASP | 0.522 (0.018) | 865.53Kb (17.19Kb) | ×2.26 |
| HalfCheetah | PPOcont | ASC | 0.160 (0.015) | 183.11Mb (0b) | |
| | | GRASP | 0.191 (0.017) | 726.18Kb (23.70Kb) | ×258.20 |
| HalfCheetah | DDPG | ASC | 34.981 (6.829) | 183.11Mb (0b) | |
| | | GRASP | 52.894 (9.820) | 4.81Mb (282.35Kb) | ×38.04 |

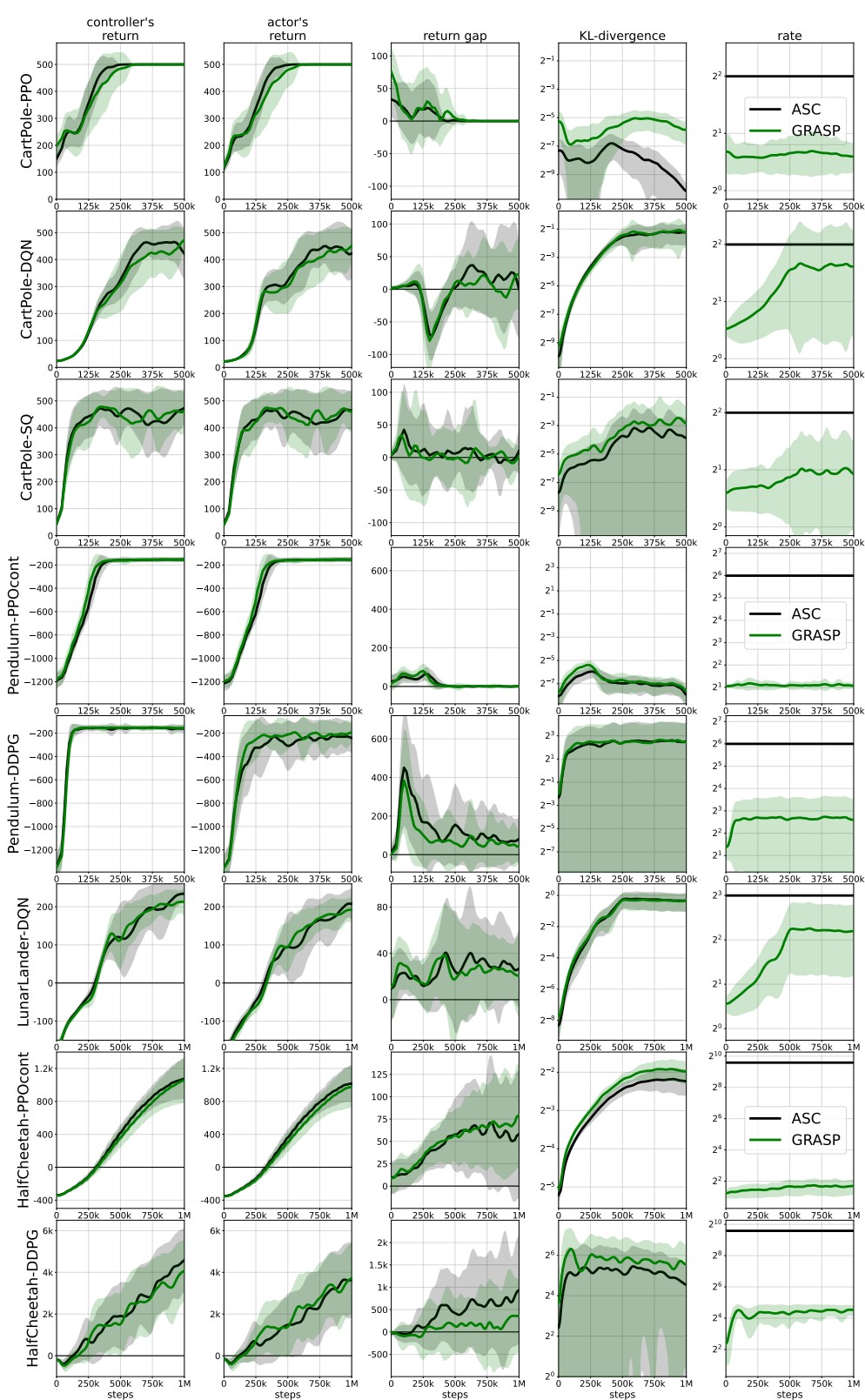

Figure 4: Training plots for different RL environments in the RRL setting.

## B    CHANNEL SIMULATION

The channel simulation method used throughout this work is Ordered Random Coding from Theis & Yosri (2022) reproduced in Algorithm 3 for convenience.

---

**Algorithm 3** GRASP Actor

---

**Require:** P, Q, N
1: $t, n, s^\star \leftarrow 0, 1, \infty$
2: $w = \min_x P(x)/Q(x)$
3: **repeat**
4:     $z \leftarrow$ sample $P$
5:     $v \leftarrow N/(N - n + 1)$
6:     $s \leftarrow t \cdot P(z)/Q(z)$
7:     **if** $s < s^\star$ **then**
8:         $s^\star \leftarrow s$
9:         $n^\star \leftarrow n$
10:    **end if**
11:    $n \leftarrow n + 1$
12: **until** $s^\star \leq t \cdot w$ **or** $n > N$
13: **return** $n^\star$

---

## C    TRAINING AND HYPERPARAMETERS

The experiments were performed on four Nvidia RTX 3080 GPUs with 10 GB of memory each, totaling 200 hours of wall clock time, including preliminary experiments. A single run of CartPole, Pendulum, LunarLander, and HalfCheetah took between 0.5 to 1.5 hours, BipedalWalker, Spread, and PistonBall took 4 to 6 hours, while Breakout and CooperativePong took 20 hours. The discount factor $\gamma$ was set to 0.99 for all environments. The hyperparameters for each

Table 5: Hyperparameter settings for PPO training in ASC and GRASP.

| env_id | total_timesteps | num_envs | learning_rate | num_steps | update_epochs | ent_coef | buffer_size | gae_lambda | clip_coef | vf_coef |
|---|---|---|---|---|---|---|---|---|---|---|
| CartPole-v1 | $5\times10^5$ | 4 | $2.5\times10^{-4}$ | 128 | 4 | 0.01 | $10^4$ | 0.95 | 0.2 | 0.5 |
| LunarLander-v2 | $10^6$ | 4 | $2.5\times10^{-4}$ | 128 | 4 | 0.01 | $10^4$ | 0.99 | 0.2 | 0.5 |
| BreakoutNoFrameskip-v4 | $10^7$ | 8 | $2.5\times10^{-4}$ | 128 | 4 | 0.01 | $10^4$ | 0.95 | 0.1 | 0.5 |
| cooperative_pong_v5 | $2\times10^7$ | 32 | $2.5\times10^{-4}$ | 128 | 4 | 0.01 | $10^4$ | 0.95 | 0.1 | 0.5 |

Table 6: Hyperparameter settings for PPOcont training in ASC and GRASP.

| env_id | total_timesteps | num_envs | learning_rate | num_steps | update_epochs | ent_coef | buffer_size | gae_lambda | clip_coef | vf_coef |
|---|---|---|---|---|---|---|---|---|---|---|
| Pendulum-v1 | $5\times10^5$ | 2 | $3\times10^{-4}$ | 2048 | 10 | 0 | $10^4$ | 0.95 | 0.2 | 0.5 |
| BipedalWalker-v3 | $10^6$ | 2 | $3\times10^{-4}$ | 2048 | 10 | 0 | $10^4$ | 0.95 | 0.2 | 0.5 |
| HalfCheetah-v4 | $10^6$ | 4 | $3\times10^{-4}$ | 2048 | 10 | 0 | $10^4$ | 0.95 | 0.2 | 0.5 |
| pistonball_v6 | $2\times10^6$ | 20 | $3\times10^{-4}$ | 2048 | 10 | 0 | $10^4$ | 0.95 | 0.1 | 0.1 |
| simple_spread_v2 | $5\times10^6$ | 3 | $3\times10^{-4}$ | 4096 | 10 | 0 | $10^4$ | 0.95 | 0.2 | 0.5 |

Table 7: Hyperparameter settings for DQN training in ASC and GRASP.

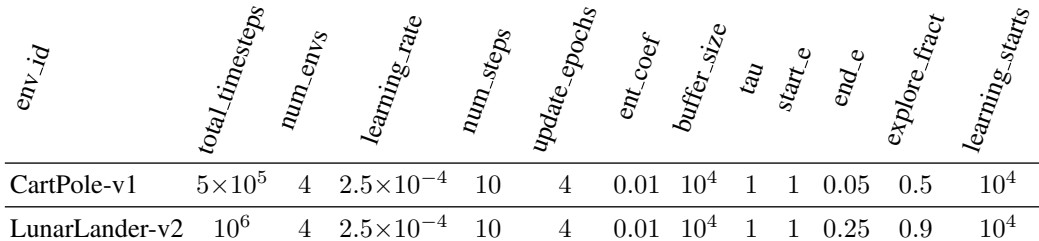

| env_id | total_timesteps | num_envs | learning_rate | num_steps | update_epochs | ent_coef | buffer_size | tau | start_e | end_e | explore_fract | learning_starts |
|---|---|---|---|---|---|---|---|---|---|---|---|---|
| CartPole-v1 | $5\times10^5$ | 4 | $2.5\times10^{-4}$ | 10 | 4 | 0.01 | $10^4$ | 1 | 1 | 0.05 | 0.5 | $10^4$ |
| LunarLander-v2 | $10^6$ | 4 | $2.5\times10^{-4}$ | 10 | 4 | 0.01 | $10^4$ | 1 | 1 | 0.05 | 0.5 | $10^4$ |

Table 8: Hyperparameter settings for SQ training in ASC and GRASP.

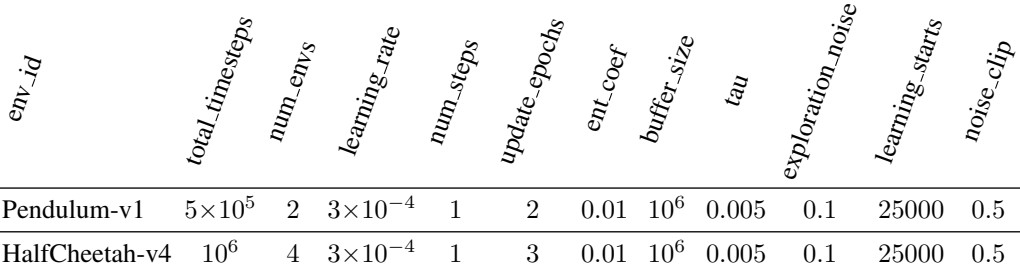

| env_id | total_timesteps | num_envs | learning_rate | num_steps | update_epochs | ent_coef | buffer_size | tau | start_e | end_e | explore_fract | learning_starts |
|---|---|---|---|---|---|---|---|---|---|---|---|---|
| CartPole-v1 | $5\times10^5$ | 4 | $2.5\times10^{-4}$ | 10 | 4 | 0.01 | $10^4$ | 1 | 1 | 0.05 | 0.5 | $10^4$ |
| LunarLander-v2 | $10^6$ | 4 | $2.5\times10^{-4}$ | 10 | 4 | 0.01 | $10^4$ | 1 | 1 | 0.25 | 0.9 | $10^4$ |

Table 9: Hyperparameter settings for DDPG training in ASC and GRASP.

| env_id | total_timesteps | num_envs | learning_rate | num_steps | update_epochs | ent_coef | buffer_size | tau | exploration_noise | learning_starts | noise_clip |
|---|---|---|---|---|---|---|---|---|---|---|---|
| Pendulum-v1 | $5\times10^5$ | 2 | $3\times10^{-4}$ | 1 | 2 | 0.01 | $10^6$ | 0.005 | 0.1 | 25000 | 0.5 |
| HalfCheetah-v4 | $10^6$ | 4 | $3\times10^{-4}$ | 1 | 3 | 0.01 | $10^6$ | 0.005 | 0.1 | 25000 | 0.5 |

