# OpenReview forum: "Remote Reinforcement Learning with Communication Constraints"
_ICLR.cc/2025/Conference — Submitted to ICLR 2025_

### Official Review · Reviewer_m2as · 2024-10-30

**Soundness:** 2
**Presentation:** 2
**Contribution:** 3
**Rating:** 3
**Confidence:** 3

**Summary:**

This paper studies remote reinforcement learning (RL) where the controller communicates with the actor through a communication constrained channel to help the actor to train a RL policy. To minimize the communication cost, the paper aims to minimize the KL-divergence between the policies of the controller and the actor. A variety of environments are used to evaluate the algorithm.

**Strengths:**

1. The problem remote RL is interesting and has practical values.
2, Information theory seems a promising tool to solve the problem.
3. The experiments are solid.

**Weaknesses:**

1. The introduction lacks of the state-of-the-art on remote RL and discussions on how this paper advances the state-of-the-art.
2. The paper presentation is informal. First, the paper does not provide the formal problem statement (minimizing the KL divergence) until Page 6. Second, lots of important details are missing in the pseudocode, e.g., channel simulation encoding, channel simulation decoding, online RL and supervised learning. These details are essential to understand how the algorithm solves the problem.
3. The key idea of the algorithm is remote sampling from information theory. However, its discussion is isolated from the algorithm. It is difficult to understand how remote sampling solves the problem of minimizing the KL divergence.
4. The experiment results (Tables 1, 2 and Figures 2, 3) do not include any classic RL algorithm, which can access the reward. The paper needs to demonstrate that the remote RL algorithm can provide comparable performance as state-of-the-art RL algorithms with lower communication overhead.

**Questions:**

Please refer to the weaknesses.

---

> ### Author Response · Authors · 2024-11-30
>
> 1. This paper introduces the novel problem of remote RL, provides canonical solutions, and demonstrates how they can be improved. To the best of our knowledge, there are no existing methods that specifically address this setting, making our work a significant contribution to this unexplored area.
>
> 2. The specifics of the channel simulation algorithm have been reproduced in Appendix B. The decoding consists of generating $n^*$-th sample from $Q$. We will ensure these details are clarified in the pseudocode for improved accessibility.
>
>
> 3. Remote sampling allows to send samples from P where the cost is $D_{KL}(P,Q)$ where $Q$ is available to both encoder and decoder. Minimizing the distance between P and Q minimizes the communication required.
>
> 3. Remote sampling allows efficient communication of samples from $P$, with a cost proportional to the KL divergence $D_{\text{KL}}(P \| Q)$, where $Q$ is known to both the encoder and decoder. By minimizing the divergence between $P$ and $Q$, the communication cost is reduced. In our setting $P = \pi_C(\cdot|s_t)$ and $Q=\pi_A(\cdot|S_t)$.
>
> 4. The ASC benchmark simulates a standard RL scenario where all information is aggregated at the controller side, effectively representing a classic RL setup.

---

### Official Review · Reviewer_PorL · 2024-10-31

**Soundness:** 1
**Presentation:** 3
**Contribution:** 1
**Rating:** 3
**Confidence:** 4

**Summary:**

This paper introduces a new scenario in reinforcement learning, termed "remote reinforcement learning," where an actor does not have direct access to the reward signal, but a controller does. Within this scenario, the authors propose a sampling approach that allows the controller to send an action with minimal communication load.

**Strengths:**

- The paper introduces a new paradigm of RL that considers practical constraints.
- Given this paradigm, the authors propose a concrete algorithm to reduce communication load while maintaining performance similar to conventional RL algorithms.

**Weaknesses:**

The main weakness of this paper lies in the motivation and rationale behind the proposed scenario. Although the authors provide detailed explanations and motivations in the introduction, I am uncertain about the practicality of the scenario.

- The limitations associated with sending reward signals might also apply to sending action signals. For instance, regarding one limitation—limited communication---while the authors differentiate between communicating actions and rewards, I question whether this distinction is significant. Rewards can be quantized, and there is extensive research on learning from noisy rewards. Quantized rewards could suffice; even in DQN for Atari games, clipped reward values perform effectively. Moreover, delayed communication could pose a problem. Communicating rewards might be feasible since rewards are mainly used for training (RL can generally handle delayed rewards), whereas delayed or noisy actions may degrade performance. Additionally, in most reinforcement learning, policies are trained in simulation and later deployed. Reward signals are typically only necessary during the training phase, while action signals are required in both simulation and deployment. I offer these examples because the practicality of this method is a central claim of the paper.
- If the primary advantage of this approach is reducing communication load, the authors should compare GRASP to a model with quantized reward signals under equivalent communication loads to strengthen their motivation.
- The first paragraph of the introduction could be improved. It states, "evaluating the reward may require solving optimization problems..." and "the challenges of lacking or costly reward acquisition in RL." However, the motivation for this scenario is related to communication, not reward acquisition challenges, as the proposed method still requires a reward signal (for the controller).

**Questions:**

See weakness

---

> ### Author Response · Authors · 2024-11-30
>
> We thank the reviewer for their insightful feedback. We are actively exploring comparisons across all proposed environments using quantized and compressed reward signals to provide a more comprehensive evaluation.
>
> In general, the rewards are real numbers they might require significant communication rate. Additionally, in multi-agent or complex environment settings, transmitting rewards alone would require each agent to learn independently, which is often suboptimal. This approach would also necessitate an additional feedback or aggregation mechanism to coordinate learning across agents, further increasing communication demands and making such system more complex.

---

### Official Review · Reviewer_HbbX · 2024-11-04

**Soundness:** 1
**Presentation:** 1
**Contribution:** 1
**Rating:** 3
**Confidence:** 3

**Summary:**

The paper introduces the novel problem of Remote Reinforcement Learning (RRL) with communication constraints, where an actor, responsible for taking actions in an environment, does not have direct access to the reward signal. Instead, a controller, which observes the reward, communicates with the actor through a constrained channel.
The authors propose a solution called Guided Remote Action Sampling Policy (GRASP), a method they claim significantly reduces communication requirements while enabling the actor to learn an optimal policy via supervised learning.

**Strengths:**

The problem is interesting. The related work appears to be comprehensive.

**Weaknesses:**

However, I find that some of the problem setup and algorithmic assumptions lack clarity and coherence, which raises foundational questions about the validity and practicality of the approach.

1. The authors assume a remote setting where the communication bandwidth is limited, yet the remote controller have access to the full state and reward signal of the environment. This setup is ambiguous: how does the controller access those info without communication? If the the controller can observe both state and reward, why it cannot access the actions as well? I suggest the authors provide a clearer justification for this selective access and an real-world example for this setup.

2. The authors assume the actor has limited computation resources, making it unable to perform RL. However, they still require the actor to perform supervised learning as shown in algorithm 2. Supervised learning can still impose non-trivial computational demands. This contradiction needs further explanation.

3. The authors assume that sending “a real number reward“ is too demanding for the communication channel (line 075), but the same channel can be used to communicate the actions. Since both actions and rewards can be continuous, it’s unclear how one can be feasible while the other is not.

**Questions:**

Apart from these foundational concerns, the paper lacks sufficient detail in the algorithmic description, the implementation, and the experiment setup. Here I list a few questions:

1. There are two copies of the actor policies being learned, one within controller learning while the one in actor learning. Are these supposed to be the same, and if so, how is consistency maintained?

2. The objective function mentioned $\pi_A$ and $\pi_C$, what do they correspond to in the algorithms?

3. What is channel simulation encoding and channel simulation decoding?

4. The author conducted experiments on MARL setting, but it’s unclear how GRASP was adapted for them.


The proposed problem is interesting, but the assumptions need more justification, and key details are missing. Addressing these issues would make the claims stronger and the paper clearer.

---

> ### Author Response · Authors · 2024-11-30
>
> Response to Weaknesses:
>
> 1. Access to information does not necessarily mean bandwidth usage. For example, a robot might know its position via GPS, while the controller has access to stationary video data or sensors. Furthermore, bandwidth constraints are often asymmetric; a base station connected to the power grid can communicate more efficiently compared to a battery-powered sensor or satellite. In this work, we analyzed the case where both the controller and the agent have access to full observations. However, modifying the information available to either the controller or the agent is an interesting direction for future research.
>
> 2. We agree with the reviewer that supervised learning can impose computational demands. In our experiments, we used the same architecture for simplicity. However, exploring smaller network architectures or alternative methods for action probability estimation is a natural next step in this line of research.
>
> 3. In general, in an MDP formulation, rewards as real numbers. In practice, they are represented as floating-point numbers. Communicating stochastic actions in GRASP via channel simulation is possible since the action is not chosen directly by the controller but it's a random outcome of the channel simulation algorithm.
>
> ---
>
> Response to Questions:
>
> 1. Both the controller and the actor learn the actor's policy based on the same set of actions and states. This process is deterministic, ensuring that the learned policies remain consistent and identical.
>
> 2. The policies $\pi_A$ and $\pi_C$ represent the actor's and controller's policies, respectively. In Algorithm 1 and 2, $P = \pi_C(\cdot|s_t), Q=\pi_A(\cdot|S_t)$.
>
> 3. Channel simulation encoding is a randomized function that maps from pair of distribution $P,Q$ to a sting of bits.
> Channel simulation decoding is a randomized function that maps $Q$ and string of bits into a sample $x$, which looks as if it was sampled directly from $P$.
>
> 3. Channel simulation encoding is a randomized function that maps a pair of distributions $(P, Q)$ to a string of bits. Channel simulation decoding is a randomized function that maps the reference distribution $Q$ and the string of bits back into a sample $x$, which follows distribution $P$.
>
> 4. We trained a single policy conditioned on the agent's identity and observations, allowing GRASP to adapt to MARL settings. We acknowledge that there are numerous possible ways to apply GRASP in MARL scenarios. GRASP is a general tool for solving remote RL problems which can be paired with any RL/MARL algorithm.

---

### Official Review · Reviewer_YpET · 2024-11-04

**Soundness:** 2
**Presentation:** 3
**Contribution:** 2
**Rating:** 3
**Confidence:** 4

**Summary:**

This paper addresses the novel challenge of Remote Reinforcement Learning (RRL) with communication constraints. In RRL, an actor takes actions based on state observations, while a separate controller receives the reward signals, and the two components communicate through a limited-bandwidth channel. To reduce the communication load, the authors propose the Guided Remote Action Sampling Policy (GRASP), which leverages importance sampling to select actions based on the controller’s optimal policy. By approximating the policy distribution through behavioral cloning, GRASP significantly reduces data transmission requirements. Experiments demonstrate that GRASP achieves up to a 50-fold reduction in communication for continuous action environments and shows comparable performance across single-agent, multi-agent, and parallel-agent environments.

**Strengths:**

1. The paper introduces the novel concept of remote reinforcement learning under communication constraints, a setting not extensively explored in the literature.
2. The paper is well-organized, clearly explaining the problem setting, methodology, and experimental design.
3. By addressing remote control scenarios in reinforcement learning, GRASP has practical applications in domains where agents operate under limited communication channels.

**Weaknesses:**

1. The originality of the GRASP solution is somewhat limited, as it primarily combines existing techniques without introducing significant new methods.
2. The comparison between the Action Source Coding (ASC) benchmark and GRASP may be unfair, as ASC lacks access to alternative action choices, whereas GRASP employs importance sampling to transmit diverse actions.
3. In Section 4 (Experiments), the paper mentions a second benchmark that transmits rewards directly to the actor, but results for this benchmark are missing from the tables and figures, leaving the comparison incomplete.

**Questions:**

1. Does importance sampling introduce high variance during training, and if so, to what extent does it impact performance?
2. Does importance sampling enhance training by introducing diverse behaviors? An ablation study quantifying this impact would be helpful.
3. What is the performance of the benchmark that transmits rewards directly to the actor, and how does it compare with GRASP?

---

> ### Author Response · Authors · 2024-11-30
>
> Response to Weaknesses:
>
> 1. GRASP is designed to address the unique challenges of remote RL with constrained communication, combining techniques such as (channel simulation and behavioral cloning) in a novel manner. While we acknowledge that it does not introduce entirely novel components, its contribution lies in the integration of these techniques to solve a practical and underexplored problem.
>
>
> 2. Theoretically, ASC and GRASP sample from the same policy distribution, so the diversity of actions should be identical. However, noise introduced during channel simulation slightly affects the samples but does not noticeably harm or benefit training, as demonstrated in our experiments.
>
>
> 3. The benchmark that transmits rewards directly to the actor would yield identical results to ASC since, in both cases, rewards, actions, and states are aggregated in one location—at the controller for ASC and at the actor for reward-sending. This equivalence is why results for this benchmark are not explicitly included.
>
> Response to Questions:
>
> 1. Importance sampling introduces minimal variance. It is only used to transmit samples from the policy distribution, ensuring consistency with the original action samples, and has negligible impact on training performance.
>
> 2. No, importance sampling is a tool to transmit samples efficiently at very low bit rates. Its purpose is to enable remote RL with limited communication without altering or enhancing the underlying training process.
>
> 3. As stated in Weakness 3, the performance of the reward-sending benchmark is identical to ASC, as both approaches aggregate information in a single location

---

> > ### Comment · Reviewer_YpET · 2024-12-03
> >
> > Dear authors,
> >
> > Thank you very much for your response; your explanation helped me understand your work. However, I will stick to my score due to the limited novelty and insufficient theoretical analysis. Thank you again for your efforts.

---

### Official Review · Reviewer_YT1F · 2024-11-07

**Soundness:** 2
**Presentation:** 2
**Contribution:** 2
**Rating:** 3
**Confidence:** 3

**Summary:**

This paper addresses the problem of communication constraints when the actor cannot access the reward signal in a timely or direct manner. To reduce communication load, the authors propose a solution that, instead of sending the actual action over the controller, employs an importance sampling method. This method samples from the desired policy to effectively learn the optimal policy.

**Strengths:**

This paper considers a common challenge in real-world applications: an agent may not receive prompt feedback when taking action.

**Weaknesses:**

1. The reward signals in real-world settings are generally less complex than the states or observations, making it potentially unnecessary to use a controller solely for receiving reward signals. Additionally, how does this approach differ from centralized training with decentralized execution? The proposed setup seems too limited in scope.
2. The controller resembles a central critic, and I don't see a significant difference between the actor-critic framework and the proposed approach.
3. Using imitation learning to train actors that mimic the controller's policies is not novel, and the proposed method lacks theoretical guarantees, relying primarily on heuristic design.
4. There is no comparison with other imitation learning algorithms such as VIPER (Verifiable Reinforcement Learning via Policy Extraction, NeurIPS 2018) or DAGGER (A Reduction of Imitation Learning and Structured Prediction to No-Regret Online Learning, AISTATS 2011).
5. The paper claims to address challenges in MARL, yet there are no MARL settings presented.

**Questions:**

See the weakness section.

---

> ### Author Response · Authors · 2024-11-30
>
> 1-2. Centralized training with decentralized execution (CTDE) is a method of training MARL systems which addresses the non-stationarity of MA-POMDPs. Unlike GRASP, CTDE does not solve the remote RL problem. However CTDE can be used in GRASP as it centralizes knowledge at the controller.
>
> 3. We agree with the reviewer, we are unaware of any guarantees for deep RL, and this remains an open challenge.
>
> 4. We introduce the remote RL problem and the GRASP algorithm, which is compatible with any RL problem. Our experiments focused on PPO, ContinuousPPO, DDPG, and DQN as baselines. A comparison with imitation learning algorithms such as VIPER or DAGGER is an interesting direction for future work, however GRASP is not an imitation learning algorithm.
>
> 5. The MARL settings CooperativePong, PistonBall and Spread are presented in Table 1 and Figure 3.

---

### Comment · Area_Chair_to9F · 2024-11-24
**From AC.**

Dear authors,

If possible, please provide a rebuttal to kick-start the discussion about the paper.

Thanks,

AC

---

### Meta-Review · Area_Chair_to9F · 2024-12-20

**Metareview:**

The paper addresses a variant of remote reinforcement learning where the actor has access to the full observation but not the reward (in the sense that the reward is only accessible via a capacity-constrained communications channel).

The main strength lies in the problem being tackled, which is interesting.

The paper has significant weaknesses.
- Transmitting a reward seems like a simpler problem than transmitting an observation / action. The paper seems to solve the simple problem but leaves the harder problem open.
- Actor requires supervised learning (which isn't always cheap).
- There are concerns about novelty / originality.

For these reasons, I recommend rejection.

**Additional Comments On Reviewer Discussion:**

All reviewers are of the opinion this is a strong reject. Author response is poor-quality and they don't address the reviews fully.
I tried to get additional feedback from the reviewers in the discussion phase to help the authros, but didn't get any replies.

---

### Decision · Program_Chairs · 2025-01-22

Reject